# Introducing the Monitoring Equipment Mask Environment

**DOI:** 10.3390/s22176365

**Published:** 2022-08-24

**Authors:** Andrea Pazienza, Daniele Monte

**Affiliations:** A3K S.R.L., 70032 Bitonto, Italy

**Keywords:** wearables, Internet of Things, artificial intelligence, edge computing, 3D printing, healthcare, safety, Medicine 4.0, medical device, bioelectronics, COVID-19

## Abstract

Filter face masks are Respiratory Protective Equipment designed to protect the wearer from various hazards, suit various health situations, and match the specific requirements of the wearer. Current traditional face masks have several limitations. In this paper, we present (ME)${}^{2}$, the Monitoring Equipment Mask Environment: an innovative reusable 3D-printed eco-sustainable mask with an interchangeable filter. (ME)${}^{2}$ is equipped with multiple vital sensors on board, connected to a system-on-a-chip micro-controller with computational capabilities, Bluetooth communication, and a rechargeable battery that allows continuous monitoring of the wearer’s vital signs. It monitors body temperature, heart rate, and oxygen saturation in a non-invasive, strategically positioned way. (ME)${}^{2}$ is accompanied by a mobile application that provides users’ health information. Furthermore, through Edge Computing Artificial Intelligence (Edge AI) modules, it is possible to detect an abnormal and early symptoms linked to possible pathologies, possibly linked to the respiratory or cardiovascular tract, and therefore perform predictive analysis, launch alerts, and recommendations. To validate the feasibility of embedded in-app Edge AI modules, we tested a machine learning model able to distinguish COVID-19 versus seasonal influenza using only vital signs. By generating new synthetic data, we confirm the highly reliable performances of such a model, with an accuracy of 94.80%.

## 1. Introduction

Personal protective equipment (PPE) is equipment worn to minimize exposure to hazards that cause serious injuries and illnesses. Respiratory Protective Equipment (RPE) is a particular type of PPE, used to protect the individual wearer against the inhalation of hazardous substances in the air. The use of medical masks and respirators as RPE is pivotal to reducing the level of biological hazard to which people are exposed during the outbreak of highly diffusible pathogens, such as the recent novel coronavirus SARS-CoV-2 [1].

Given the continuing health emergency due to the SARS-CoV-2 coronavirus pandemic, face masks have become a commonly used tool to prevent the transmission of COVID-19 infection and other respiratory viruses. Over time, awareness of the effectiveness of wearing face masks as an individual protective device has matured [2].

The use of the mask concerns only one of the various prevention and control measures that can limit the spread of some respiratory viral diseases, including COVID-19. Masks generally perform several functions, and are recommended to be the last level of defense to: (i) prevent the spread of germs and infectious diseases including COVID-19; (ii) protect the wearer from excessive chemical exposure; (iii) prevent unnecessary injury; and (iv) help businesses comply with regulatory requirements. A mask not worn or incorrectly worn is totally ineffective and may give the user a false sense of protection.

However, traditional masks, whether surgical or filtering (FFP2 or FFP3), have many limitations. Surgical masks, in fact, do not protect the wearer from microscopic infectious particles such as viruses. The masks equipped with filters, in turn, favor the exhalation of air, which, being released to the outside through a filter, makes them risky, as they protect the wearer, but not the people with whom they come into contact. Additionally, since the beginning of the COVID-19 pandemic, surgical or filtering masks were used worldwide, which, for the most part, were usually of the disposable type and have therefore been disposed of up to now, recalling the consumer to the continuous and demanding call to a responsible and sensitive use of waste management, sustainable ecology and respect for nature.

Interestingly, given the inherent ability of masks to be wearable, they turn out to be a tweaked active form of wearable devices for healthcare. Healthcare wearables enable the remote monitoring of vital signs and health statistics. Patients gain better visibility into their health, resulting in improved treatment outcomes. Wearable medical devices enable providers to deliver better care, improve efficiency, and reduce operating costs. Internet of Things (IoT) and smart connected technologies, in combination with Artificial Intelligence (AI) applications, can play a critical role not only in the prevention, mitigation, and continuous remote monitoring of patients but also in achieving a prompt response in cases of danger or when detecting a possible early stage of pathology.

IoT-enabled medical wearable devices provide individuals with the information needed to achieve better health outcomes. Healthcare wearables leverage non-invasive integrated sensors into relevant aspects of an individual’s health status, including (i) real-time health monitoring; (ii) alerts and alarms to monitor vulnerable patients’ health status; and (iii) patient–physician information sharing.

Therefore, in this paper we introduce (ME)${}^{2}$, the Monitoring Equipment Mask Environment, which is an evolution of an RPE with enhanced functionalities. The present innovative mask concerns: (1) a 3D-printed reusable mask with eco-sustainable materials and replaceable filter pack; (2) sensing modules to collect vital parameters; (3) a microcontroller equipped with a microphone and a rechargeable battery; (4) an associated mobile application which provides information about the health status; and (5) one (or more) in-app on-edge Machine Learning (ML) module(s) to perform predictive data-driven analysis, acting as an early warning system.

To address these requirements, we designed, developed, and produced a prototype of the (ME)${}^{2}$ innovative mask, depicted in Figure 1. Summarizing the efforts, the main contributions of our work are the following:Protection: we ensure the protective aims of the mask from breathing in substances hazardous to health and enhance the protection by introducing antimicrobial materials onto the 3D-printed mask, capable of reducing the viral and bacterial load on the surface of the mask within a few minutes of exposure.Eco-sustainability: there are a few benefits of 3D printing when it comes to sustainability, including the fact that (i) processes use far less energy and produce fewer emissions in the printing process; (ii) materials used are biodegradable and renewable; (iii) 3D printing can prolong the use of older equipment by creating spare parts for just about anything, including furniture and equipment; (iv) printing can be done on demand while addressing supply chain challenges; and (v) local 3D printing services reduce the need for shipping.Customization: The digitization of the (ME)${}^{2}$ 3D model design allows us to introduce the benefits of parametric modeling. The parametric modeling process allows for intent and relationships to be created between geometric features, which means the shape of your model changes as soon as a dimension value is modified. In this way, we can design a mask that fits perfectly on the face of everyone. Moreover, for the sake of customization, we may include specific branding on the mask.Connection: We studied and designed sensing targets embedded into a face mask, identifying and locating the best vital sensors that non-invasively collect vital signs on a novel part of the face, namely on the nose, while maintaining the weight balancing of the mask. This is a major contribution to the originality of our work. The current vital signs captured are body temperature (${}^{{}^{\circ}}$C), oxygen saturation (SpO${}_{2}$), and heart rate (beats per minute, bpm). We also designed an energy-efficient run-time architecture with a long battery lifetime handling, an on-device signal processing pipeline, and we managed to convey gathered information to the mobile application via Bluetooth Low Energy (BLE).Intelligence: To prove the validity of data analytics tasks, we tested a supervised ML model that distinguishes patients with influenza and SARS-CoV-2 infections using the available vital signs. The ML model runs on-edge in the mobile application, ensuring the privacy of sensitive data, and supporting the user’s awareness of his health status with visualization tools and alerts.Innovation: the use of innovative technologies such as the IoT and AI creates a combination of digital solutions that “come to life” with the production of digital manufacturing prototypes in 3D printing.

This paper is organized as follows. Section 2 provides the background. Section 3 presents the (ME)${}^{2}$ innovative mask with the key concepts of design, sensing, implementation, and AI modelling. Section 4 describes the results of applying AI model to differentiate COVID-19 symptoms from seasonal influenza. Section 5 discusses the main contributions of this paper with state of the art in the literature. Finally, Section 6 concludes the paper, outlining future perspectives.

## 2. Background and Related Work

RPE includes full-face respirators, self-contained breathing apparatus, gas masks, N95 respirators, and surgical masks that are used for a task that can cause inhalation of harmful materials to enter the body. This includes harmful gas, chemicals, large-particle droplets, sprays, splashes, or splatter that may contain viruses and bacteria such as COVID-19, viral infections, and more.

IoT is continuing to gain popularity, particularly in the sphere of healthcare. The benefits of IoT wearable devices for healthcare are numerous:Transitioning from reactive to predictive medicine;Simplify physiological data collection;Monitor the intensity of human physical activities;Improves medical treatments;Collect real-time patient data with minimal to no inconvenience or discomfort;Receive alert notifications for early detection of diseases and adverse health events;Increase patient engagement by giving patients more control over their treatment;Reduce costs for both healthcare providers and patients;Proactively manage chronic conditions.

Healthcare technologies using IoT, especially wearable technology, can help medical practitioners follow IoT trends in healthcare and also optimize remote monitoring care for the global population. With IoT in the healthcare industry, care providers can automate regular analyzing processes, and are able to pay more attention to the important things that require human intervention.

Wearables collect large amounts of data that help the physician determine the correlation between health conditions to be dealt with efficiently. A wearable medical device can collect data on physiological and biochemical parameters, which is crucial in cases of patients who need to track a specific physical activity [3,4].

Smart healthcare using IoT can even help simplify diagnosing [5,6,7]. By allowing patients to self-diagnose their own health conditions, IoT healthcare solutions can help hospitals reduce the costs associated with their care [8,9]. IoT in healthcare can provide the right insight to make more justified decisions about a particular treatment based on patient data collected. It can also reduce the healthcare costs spent on payments for additional visits to the healthcare providers, as well as costs for on-site healthcare services. With the ability to diagnose patients remotely, healthcare will be more optimized and responsive.

The complexity and rise of data in healthcare means that AI will increasingly be applied within the field. Several types of AI are already being employed by payers and providers of care, and life sciences companies [10,11,12]. In the form of machine learning (ML) and deep learning (DL), it is the primary capability behind the development of precision medicine, widely agreed to be a sorely needed advance in care.

However, since the beginning of the COVID-19 pandemic, academic researchers and the industry sector have been interested in expanding the purpose of masks from protection to comfort and health, leading to the conceptualization or the release of various “smart” mask products around the world. While most smart masks presented in the market focus on resolving problems with user breathing discomfort, which arise from prolonged use, academic prototypes were designed for not only sensing COVID-19 but also general health monitoring aspects. In this context, an interesting recent review is conducted in [13].

In the recent literature, we highlight some similar works that may result in having some common features with our solution. A detailed comparison of existing masks with the proposed innovative mask is shown in Table 1.

In particular, [14] presented the “Smart Mask” concept, i.e., a digitized wearable protection that will provide improved protection against infections spreading through aerosol or respiratory droplets with integrated real-time data analysis for improving the immediate safety of the users. In addition to protecting users, the Smart Mask would collect the health and health-related data of the user, offering new tools for predictive and preventive healthcare.

In the same spirit, [15] presented “FaceBit”, i.e., an open-source research platform for smart face mask applications, accompanied by a mobile application that provides a user interface and facilitates research. It monitors heart rate without skin contact via ballistocardiography, respiration rate via temperature changes, and mask-fit and wear time from pressure signals, all on-device with an energy-efficient run-time system.

Other works [16,22] proposed “ADAPT”, a smart IoT-enabled active mask. This wearable device contains a real-time closed-loop control system that senses airborne particles of different sizes near the mask by using an on-board particulate matter (PM) sensor.

In [17], a lightweight and zero-power smart face mask is introduced, capable of wirelessly monitoring coughs in real time and identifying proper mask wearing in public places during a pandemic.

Furthermore, works in [18,23] presented a smart mask that allows the monitoring of body temperature and breathing rate. Body temperature is measured by a non-invasive dual-heat-flux system, consisting of four sensors separated from each other with an insulating material. Breathing rate is obtained from the temperature changes within the mask, measured with a thermistor located near the nose. The system communicates by means of long-range (LoRa) backscattering, leading to a reduction in average power consumption.

The authors of [19] presented a prototype of a smart face mask, the “AG47-SmartMask”. In addition to having the function of both an active and passive anti-COVID-19 filter, the AG47-SmartMask also allows the continuous monitoring of numerous cardio-pulmonary variables. Several specific sensors are incorporated into the mask in an original way that assess the inside-mask temperature, relative humidity and air pressure together with the auricular assessment of body temperature, heart rate and oxygen saturation.

In [20], a technology-garment co-designed smart mask concept is presented, Masquare, to enable the use of daily face garments with cardio-respiratory health monitoring functions.

Finally, Ref. [21] designed a smart mask that carries out respiratory rate monitoring, based on the difference in the temperature of the breathing, which was read by the temperature sensor and displayed in the blynk application on the smartphone.

## 3. Materials and Methods

In the following subsections we detail the (ME)${}^{2}$ innovative mask concept and the main components including the Mask Design and Materials (Section 3.1), the sensing modules and IoT components to capture body temperature, heart rate, and oxygen saturation (Section 3.2), the mobile app implementation (Section 3.3), and the AI and ML data analytics (Section 3.4).

### 3.1. Mask Design and Materials

The (ME)${}^{2}$ innovative prototype consists of a 3D-printed filter mask equipped with electronics and sensors in order to continuously and non-invasively monitor the vital parameters of the wearer. The 3D printing of the (ME)${}^{2}$ innovative mask takes place by means of the additive manufacturing technique, starting from a digital 3D model of the mask and its components. The 3D digital model is created with parametric modeling, which guarantees the aspects of uniqueness, comfort and ergonomics, as well as high precision in the three-dimensional configuration of the mask. This design feature ensures optimal adhesion, allowing the printing technique to perfectly conform the mask to the user’s face.

The (ME)${}^{2}$ 3D model is made up of several parts, which are printed in two different types of filaments. The mask part is printed in filaments of thermoplastic polyurethane (TPU), which is an elastic and flexible material with excellent resistance to abrasion and cutting, enabling a more comfortable mask when worn compared to rigid mask parts.

Additionally, the nose-piece, the holder cover, the filter cover, and the head clip are printed in rigid polylactide (PLA) filaments, which is a bio-plastic of natural origin derived from plant substances, in particular from corn starch; it is therefore a totally biodegradable material and is water-soluble at temperatures above about $75{\;}^{{}^{\circ}}$C. Being a material of natural origin and, therefore, non-toxic, PLA is compatible to come into contact with the face.

Specifically, we exploited the antimicrobial material from Copper3D (https://copper3d.com/, accessed on 21 August 2022). The properties of this material are suitable for the (ME)${}^{2}$ innovative mask, as it can reduce the action and eliminate the 99.975% of fungi, viruses, bacteria, and a wide range of microorganisms (including coronavirus) in less than one hour (90% in five minutes) [24,25]. In particular, MDFlex^®^ TPU98A and PLACTIVE^®^ AN1 nanocomposites have been employed.

Hence, the idea was to solve all the problems related not only to the correct adherence to the face of the mask but also to its perfect bio-compatibility with the epithelial tissue of the human face and to the protection of the respiratory tract, aspects that an RPE must necessarily guarantee.

The 3D-printed elements were designed with Rhinoceros (https://www.rhino3d.com/, accessed on 21 August 2022). Figure 2 presents the designed mask component for 3D-printing. The (ME)${}^{2}$ innovative mask has been printed with the Raise3D Pro2 Plus 3D printer. Currently, the mask weight is 89 g (the electronics compartment is 39 g and the mask without the electronics compartment is 50 g).

Therefore, the (ME)${}^{2}$ innovative mask evolves from a mere individual protection tool to a more complex digital health device, including information and communication technologies to support and promote disease prevention, diagnosis, treatment and monitoring, health management, and lifestyle. To this end, the 3D model of the (ME)${}^{2}$ innovative mask is adequate to contain the electronic components and the correct positioning of the same to ensure a weight balance of the mask on the face.

### 3.2. Sensing and IoT Modules

Continuous monitoring is of the utmost importance for the management of health and safety emergencies. The aim of the sensing and IoT module is to provide users with accurate and reliable health information. Therefore, in this work, we focused on the types of vital sensors that need to be integrated into the prototype of the (ME)${}^{2}$ innovative mask to obtain such data. For instance, it has been reported that by following peripheral oxygen levels (SpO${}_{2}$), heart rates (HR), and body temperature (T), medical staff can prevent the aggravation of the symptoms of respiratory diseases, such as the one caused by the severe acute respiratory syndrome coronavirus (SARS-CoV-2).

As shown in Figure 3, the (ME)${}^{2}$ innovative mask prototype employed:MAX30102: a commercial photometric module to capture SpO${}_{2}$ and HR (Figure 3a);BlueDot TMP117: a commercial body temperature sensor (Figure 3b);LILYGO^®^ TTGO T-Display ESP32: a low-power system-on-a-chip (SoC) micro-controller (Figure 3c);INMP441: a commercial microphone for single-board computers (SBCs) (Figure 3d).

Regarding SpO${}_{2}$ and HR levels in diseases, respiratory impairment it has been shown to reduce oxygen saturation [26], while feedback mechanisms enhance the cardiac frequency by positive chronotropic effect [27]. This condition is known in many infectious respiratory diseases, such as SARS-CoV-2, but is not limited to them. It is widely reported in the literature that declining SpO${}_{2}$ and ventricular tachycardia are a strong indication of chronic obstructive pulmonary disease as well as other cardiopulmonary and circulatory disorders [28].

Monitoring of SpO${}_{2}$ and HR in the clinical setting is performed by pulse oximeters, which evaluate saturation through photometric means [29]. In this sense, the probe of the device houses two light-emitting diodes (LEDs), which emit light at 660 and 940 nm, as well as a photodiode [30]. This system is typically placed on the user’s finger in order to create an interface between the LEDs and the photodiode so that the light intensity captured by the photodiode changes according to the concentration of oxygen in the blood and due to the passage of blood through the finger. The resultant photoplethysmogram allows both the mensuration of SpO${}_{2}$ and HR [31]. Furthermore, at a particular light intensity when the available light is below a threshold value, it can be assumed that the mask does not adhere well to the user’s skin and, consequently, does not wear on the user’s face, giving us the possibility to detect whether the mask is worn or not.

The MAX30102 includes internal LEDs, photodetectors, optical elements, and low-noise electronics with ambient light rejection. The MAX30102 provides a complete system solution to ease the design-in process for mobile and wearable devices. The sensor operates on a single 1.8 V power supply and a separate 3.3 V power supply for the internal LEDs. Communication is through a standard I${}^{2}$C (Inter-Integrated Circuit) compatible interface. The module can be shut down through software with zero standby current, allowing the power rails to remain powered at all times.

The BlueDot TMP117 (specifically, the TMP117NAIDRVR) is a high-precision temperature sensor from Texas Instruments, capable of measuring temperatures with an accuracy of up to $\pm 0.1{\;}^{{}^{\circ}}$C across the range of $-20{\;}^{{}^{\circ}}$C and $50{\;}^{{}^{\circ}}$C with no calibration. The sensor operates from 1.8 V and 5.5 V at VCC, dispensing the need for voltage regulation. It has a very low power consumption, typically 3.5 μA, which minimizes the impact of self-heating. The sensor communicates through the I${}^{2}$C protocol using 0 × 49 as the default address. In order to perform weight balancing, the TMP117 has been positioned in a specular way on the other side of the nose. Both the MAX30102 and TMP117 sensors are inserted into a 3D-printed nose clip, whose 3D parametric model ensures the perfect adherence to any type of nose.

The INMP441 is a 3.3 V MEMS microphone that uses Inter-IC Sound (I${}^{2}$S) to communicate with devices capable of audio recording via the I${}^{2}$S interface. The INMP441 is introduced for voice inputs, sound localization, and other applications where an array of microphones can be used to identify and characterize acoustic systems.

The LILYGO^®^ TTGO T-Display ESP32 is basically an ESP32 development board with some added hardware features, that is, a color LCD display, a battery charging interface, two onboard GPIO buttons and a USB-C connector. Specifically, the ESP32 has a chipset ESPRESSIF-ESP32 240 MHz Xtensa^®^ single-/dual-core 32-bit LX6 microprocessor, with 4 MB QSPI flash and 520 kB SRAM, and Wi-Fi, Bluetooth v4.2, and BLE standard connections.

The overall operation of the device consisted of the signal acquisition from the MAX30102 and TMP117 sensors, and the INMP441 microphone. These signals were then transmitted to the ESP32 by I${}^{2}$C and I${}^{2}$S protocols. Moreover, the MAX30102 and TMP117 sensors were powered by the ESP32. The firmware in ESP32 SoC then allowed the information to be shown on the display, as well as to use ESP32 IoT capabilities to transmit data via BLE. The firmware was developed using Visual Studio Code with the Platform.io (https://platform.io/, accessed on 21 August 2022) extension. Both native and external libraries were used to better communicate with MAX30102. Some specific functions have been implemented directly with low-lever registries to enable the deep-sleep functions of the board. The assembly of the device consisted of integrating both the sensing modules and ESP32 SoC, specifically connecting MAX30102, TMP117, and INMP441 to the ESP32 with copper connection cables that are soldered on the board using the appropriate pins.

Energy management is an important aspect of the (ME)${}^{2}$ innovative mask. A LiPO battery has been adopted, with 3.7 V voltage, 1100 maH of capacity for 4.07 Wh, for a weight of 19 g. With the monitor always in continuous use, the 8-h battery life is measured and optimized to turn off the monitor when not needed, while, when deep sleep mode is set, the battery life lasts up to 4 weeks. Lastly, a JST 1.25 connector has been used to connect to the board without the need for soldering or adapters.

In summary, we considered various aspects of sensor and wiring design and electronics placement in the (ME)${}^{2}$ innovative mask. Primarily, we empirically identified the best point on the face to be in contact with the sensors, as the type and location of those sensors have an impact on measurement results. In this way, we guarantee real-time reliable communication to inform users of their health status. Accordingly, we delved into optimizing the battery life to guarantee full protection when the (ME)${}^{2}$ innovative mask is worn in the workplace. Finally, Figure 4 shows a graphic diagram of all designed 3D components and the electronics of the (ME)${}^{2}$ innovative mask, in order to have a clearer view of the big picture.

### 3.3. Associated Mobile Application

The (ME)${}^{2}$ innovative mask communicates via BLE to an associated mobile application (in short, (ME)${}^{2}$ app). The rationale behind the (ME)${}^{2}$ app is to report a series of useful information about the mask and the user’s health information. The essential information shown on the screen are as follows:Check the connection established with the (ME)${}^{2}$ innovative mask;Indicate if the mask is worn;Indicate how long the mask has been worn;Indicate if it is necessary to change the filter;Show real-time health status:
Heart rate (in beats per minute);Blood oxygen saturation (SpO${}_{2}$, in percentage);Body temperature (in ${}^{{}^{\circ}}$C).

The (ME)${}^{2}$ app acts as a client for the BLE server on the (ME)${}^{2}$ innovative mask, connecting to it after associating it through the appropriate procedure. Once the name and MAC address have been retrieved, this is stored and used for subsequent connections. In particular, if the app is running, it reads the data using the BLE properties every 10 s; otherwise, a background service subscribes to the characteristics which are updated from the mask.

The (ME)${}^{2}$ app is developed for Android and iOS using Flutter with the Dart programming language (https://flutter.dev/ accessed on 21 August 2022) Data are stored in a local sqflite database. Envisioning a full personal health device, the (ME)${}^{2}$ app also accepts further user general and health information, some of which are requested to be inserted only at the first time the app is accessed, while others can be updated accordingly, in order to manage a comprehensive user profile. At the current version of the app, the following information are requested:Sex;Age;Weight (in Kg);Height (in cm);Blood pressure (both Systolic and Diastolic, in mmHg);Respiratory rate (in breaths per minute).

In particular, from weight and height, other useful information are derived, such as the Body Mass Index (BMI), and the Body Surface Area (BSA).

Figure 5 depicts the application’s home screen, health sensor device details, and other health data inserted by the user. The homepage is a dashboard for at-a-glance information about the user’s health status. Notifications can be displayed or pushed to the user.

The role of the (ME)${}^{2}$ app is to bridge the gap between wearable medical devices and mobile health technologies (mHealth). Alerts and alarms can be promptly notified about the deterioration of the user’s clinical conditions. Additionally, recommendations and reminders in the form of notifications can be set to act as prevention actions. In the event of a detected danger and no response on the app from the user, we developed the possibility of contacting the physician directly, sending them a health status report, or, in the worst case, warning the emergency health personnel with contextual information, such as the geo-localized position.

### 3.4. Edge AI

In this scenario, AI techniques can play a crucial role. The development of AI-based techniques may support (ME)${}^{2}$ users to cope with the changes in their health status and healthcare assistance. Due to various challenging issues such as computational complexity and the increased delay times in Cloud Computing, Edge Computing is an emerging paradigm and a promising solution that pushes computing tasks and services from the network core to the network edge. Recently, Edge Computing has overtaken the conventional process by efficiently and fairly allocating the resources, i.e., power and battery lifetime in IoT-based healthcare applications. In the meantime, considering that AI is functionally necessary for quickly analyzing vast volumes of data and extracting insights, there exists a solid demand for integrating Edge Computing and AI, which gives the birth of Edge AI. Taking automated complex decisions with the help AI-based techniques directly on the Edge enables a faster, more private, and context-aware Edge Computing empowering. As a result, the physical proximity between the computing and information-generation sources promises several benefits compared to the traditional cloud-based computing paradigm, including low latency, energy efficiency, privacy protection, reduced bandwidth consumption, and context awareness.

Therefore, the (ME)${}^{2}$ innovative mask goes in this direction by including suitable supervised ML models in the (ME)${}^{2}$ app to predict early diagnoses. Several ML tasks may be embedded into the (ME)${}^{2}$ innovative mask, acting as an on-edge medical device. Data collected from vital sensors may be used to predict a level of clinical risk assessment, or pathologies related to cardiovascular or respiratory diseases. Moreover, by analyzing breaths or coughs by means of the microphone, other pulmonary diseases may be detected. As an example scenario, Figure 6 shows the AI module inside the (ME)${}^{2}$ app.

A strong peculiarity of our edge AI approach refers to the possibility to train and validate the supervised ML model in laboratory conditions in Python with classical data science tools, and then, once the model is ready to use, it can be serialized with Pickle and then transpiled in Dart native code language with the m2cgen (https://github.com/BayesWitnesses/m2cgen accessed on 21 August 2022) Python library, to be correctly run in the Flutter framework with zero dependencies.

## 4. Results

In this section, we firstly test the measurement performances, and then we evaluate the ‘intelligence’ of the (ME)${}^{2}$ innovative mask.

Since the mask covers the face, we need to find a point on the face that allows us to make accurate measurements. In our studies, we employed the MAX30102 sensor in different face and nose points, as shown in Figure 7.

Experiments have been conducted to validate the position. The data points collected were compared with those of the Beurer PO 30 pulse oximeter, placed on the finger, which are taken as ground-truth data. In particular, the latter has an accepted measurements error of ±2% for both pulse rate (±2 bpm) and SpO${}_{2}$ (70–100%). The results are shown in Table 2. It can be seen that data gathered from point 7, on the nose, perform better and can be adjusted accordingly to achieve reliability of such health information.

In order to validate the Edge AI module of the (ME)${}^{2}$ innovative mask, we employed the use case from work in [32]. Specifically, this work developed and validated a supervised ML pipeline to distinguish patients between two viral infections, particularly with seasonal influenza and COVID-19, using only available vital signs and demographic data. Vital signs are critical piece of information used in the initial triage of patients with COVID-19 and/or influenza by health coordinators and health workers in community urgent care centers or emergency rooms. Therefore, the main idea was allow for rapid and accurate triage patients, as it is becoming clearer that patient vital signs may present uniquely in SARS-CoV2 infection, possibly due to alterations.

This work has shown the capacity of machine algorithms in differentiating between COVID-19 and influenza patients using basic clinical variables. The dataset is collected from a study cohort of West Virginia University (WVU) Hospital. Although such data were not public, the trained model was available online (COVID-19 vs. Influenza: https://github.com/ynaveena/COVID-19-vs-Influenza (accessed on 21 August 2022)). Specifically, an XGBoost predictive model was trained. We transpiled this model in Dart native code and tested it with new generated synthetic tuples. Specifically, we exploited the Python module DataSynthesizer (https://github.com/DataResponsibly/DataSynthesizer, accessed on 21 August 2022), a tool that takes a sensitive dataset as input and generates a structurally and statistically similar synthetic dataset with strong privacy guarantees. Therefore, we generated a dataset with 150 tuples and 10 features of which 71 (47%) were of the COVID-19 positive class and 79 (53%) of the influenza class. Metrics performances are shown in Table 3.

Our results substantially confirm the ones from [32], which are almost comparable. This means that the XGBoost model generalizes well, specially with new (synthetic) data. Moreover, the model, transpiled in Dart native code directly from a Python serialized object, has no impact in reducing the performances.

## 5. Discussion

Through the use of the (ME)${}^{2}$ mask it is possible to predict, in a sufficiently early manner, the onset of general pathologies, such as pyrexia, or of a special pathology, linked to the respiratory and/or cardiovascular organs. One of the indisputable advantages is inherent in the fact of having conceived an innovative mask of high manufacturing simplicity through the use of the 3D printing application, which, being able to produce masks in a completely customized way, gives itself and its user unique aspects of pleasantness and eco-sustainability, maximized by the excellent hermetic seal of the mask to the user’s face.

Another advantage, deriving from the previous one, is inherent in the fact that the excellent hermetic seal of the mask prevents and completely solves the problem of the fogging of the user’s glasses.

Furthermore, the use of ecological materials used for its construction gives it the characteristics of easy washing and sanitizing action.

This device also has a further advantage, deriving from the previous one, of not producing any environmental impact, since it is made up of easily recyclable component materials. A further advantage is constituted by the fact that the present device can be industrially produced in a very wide range of applications, geometries, dimensions, and colors of the mask body, such as to satisfy the most varied fashion and market needs.

This device is therefore validly usable and represents a new opportunity for the development of the current economy applied both to devices for individual protection against biological agents such as the virus that caused the COVID-19 pandemic, and because it is still suitable for the remote and real-time monitoring of all those who wear it and, therefore, applicable to a plurality of further sectors to solve all the problems currently unresolved by known methods.

## 6. Conclusions

Beyond the limitations of traditional face masks, there is room for sensing and intelligence that could inform and actively protect wearers. Smart face masks could provide an early warning system when health or environmental metrics indicate danger. If a smart mask gathers useful health data, such as heart rate, oxygen saturation, and body temperature, one may prevent injuries, illnesses, and fatalities, so as to ensure a safe and healthy environment. Therefore, we present (ME)${}^{2}$, the innovative Monitoring Equipment Mask Environment, which integrates eco-sustainable 3D-printed materials, heath sensing, IoT, AI and an associated mobile app to help gathering health data and detect health-related events in real-time.

Several efforts are employed into the wearable and sensing design to achieve a valid trade-off between comfort, weight balancing, and the precision of sensors’ measurements.

Moreover, AI modules are embedded in the mobile app, in an on-edge fashion. As a result, the transmission latency of data offloading decreases, the data privacy increases and the WAN bandwidth cost reduces. The use of simple ML-based classification may have utility for the rapid identification, triage, and treatment. As example scenario, we exploited a publicly available model of COVID-19 and influenza-positive patients, which is especially relevant as the influenza season approaches.

Hence, the applicability of this device extends not only to anyone who needs to wear a filter mask, but also to patients, workers, and stakeholders who need continuous monitoring of vital signs in real-time and remotely.

In future works, ad hoc printed circuit boards (PCBs) optimized for the use of BLE will be chosen so as to reduce consumption in the operating state and gain battery life. Furthermore, we want to introduce new health detection capabilities, such as respiratory rate, which is currently difficult to monitor in a wearable setting.

## 7. Patents

A. Pazienza, D. Monte. Mascherina intelligente con vestibilità personalizzata, munita di un dispositivo “IoT” atto al suo monitoraggio medicale. Italian Patent IT 102022000006884. Ufficio Italiano Brevetti e Marchi (UIBM). Filed on 6 April 2022.

## Figures and Tables

**Figure 1 sensors-22-06365-f001:**
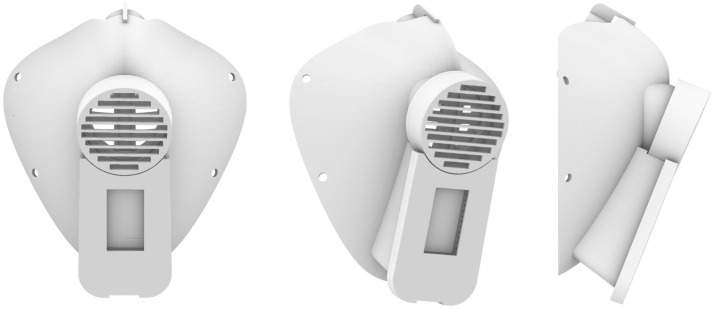
Design images of the assembly of the (ME)${}^{2}$ 3D-printed innovative mask.

**Figure 2 sensors-22-06365-f002:**
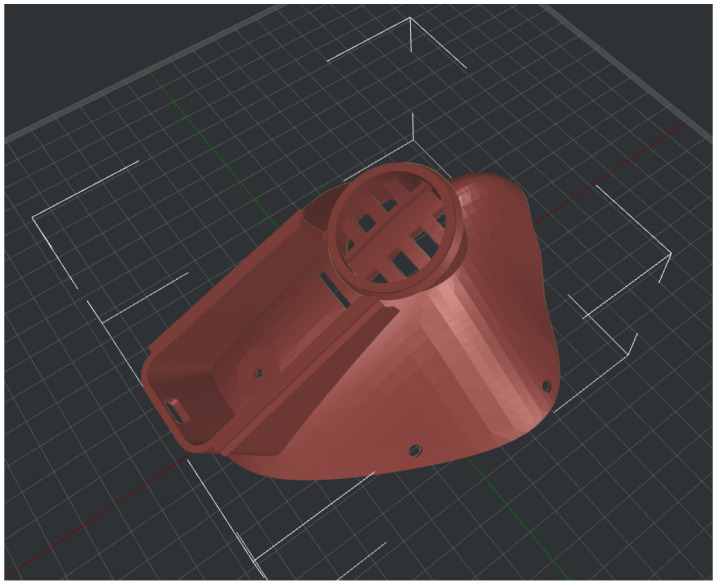
3D Model Design of (ME)${}^{2}$ innovative mask.

**Figure 3 sensors-22-06365-f003:**
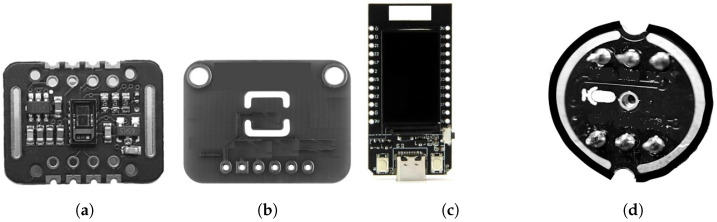
Sensors employed in the IoT Modules. (**a**) MAX30102; (**b**) BlueDot TMP117; (**c**) ESP32; (**d**) INMP441.

**Figure 4 sensors-22-06365-f004:**
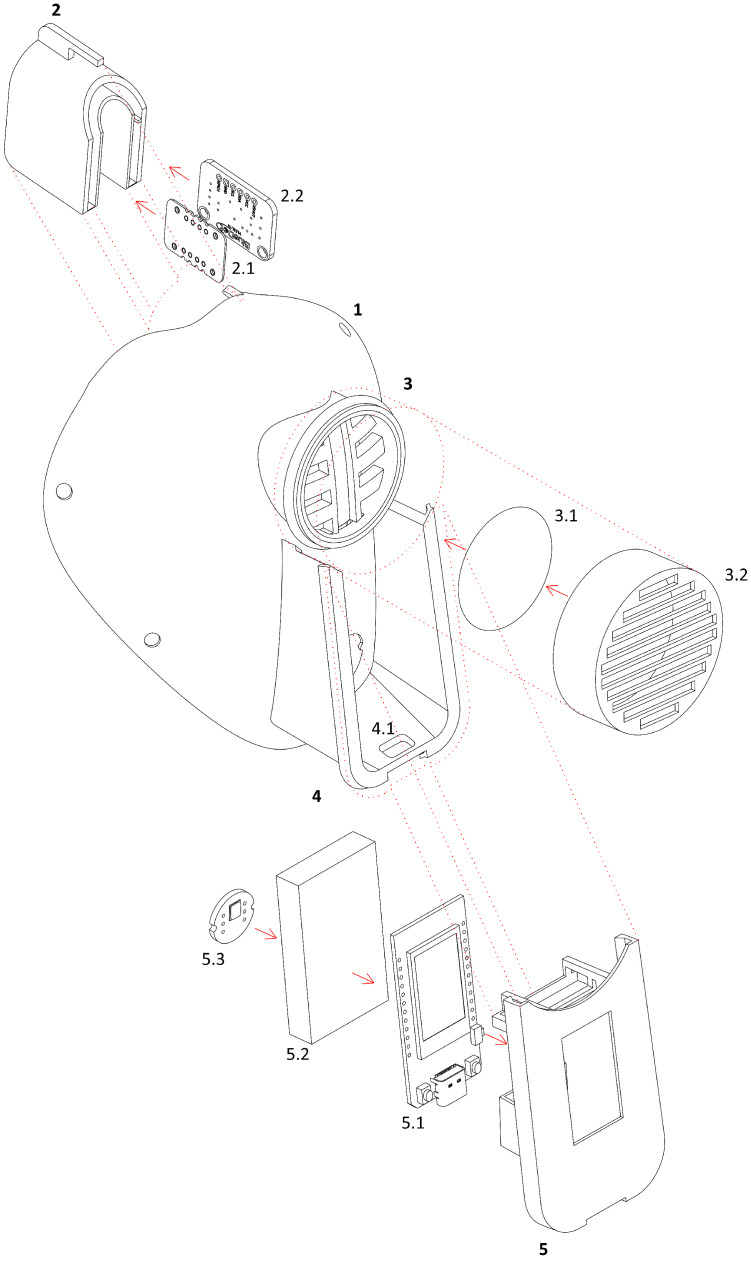
Graphic diagram showing an exploded three-dimensional enlargement of the (ME)${}^{2}$ innovative mask (1) of Figure 1, from which it is possible to deduce its component parts, in particular: the nose-piece (2) of the mask (1), whose shape is suitable for contain a pair of sensors (2.1–2.2), in the two very small separate housings obtained in the oblong parts of the same nose pad, i.e., the heart rate and the blood oxygen saturation level sensor (2.1) and the temperature sensor (2.2), respectively; the mask body 3, comprising a support of the filter (3.1), in turn protected and enclosed within a filter cover (3.2); a support and containment medium (4) of the electronic board (5.1), the latter in turn inserted in a support cover (5), which can be constrained to a corresponding base (4.1) rigidly constrained to the body of the template (1) and also containing, together with it, the lithium battery (5.2) and microphone (5.3).

**Figure 5 sensors-22-06365-f005:**
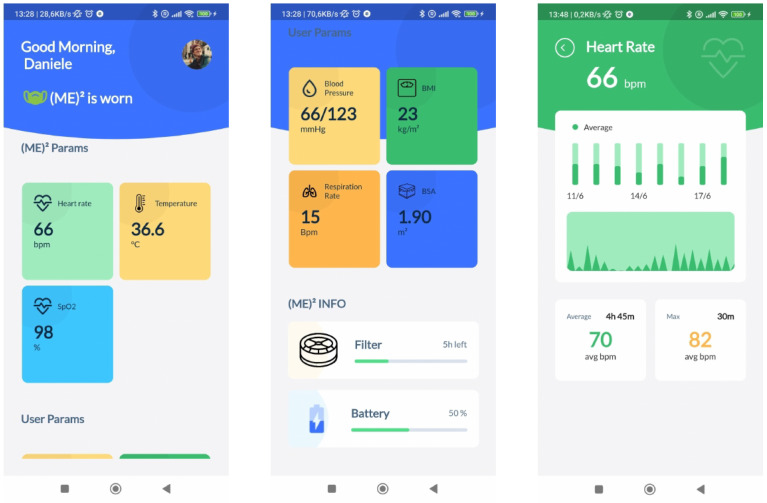
(ME)${}^{2}$ app on Android. The homepage displays general details, if the mask is worn, real-time vital signs heart rate, body temperature, and oxygen saturation, other offline health data, user general information, filter life status, and battery life status.

**Figure 6 sensors-22-06365-f006:**
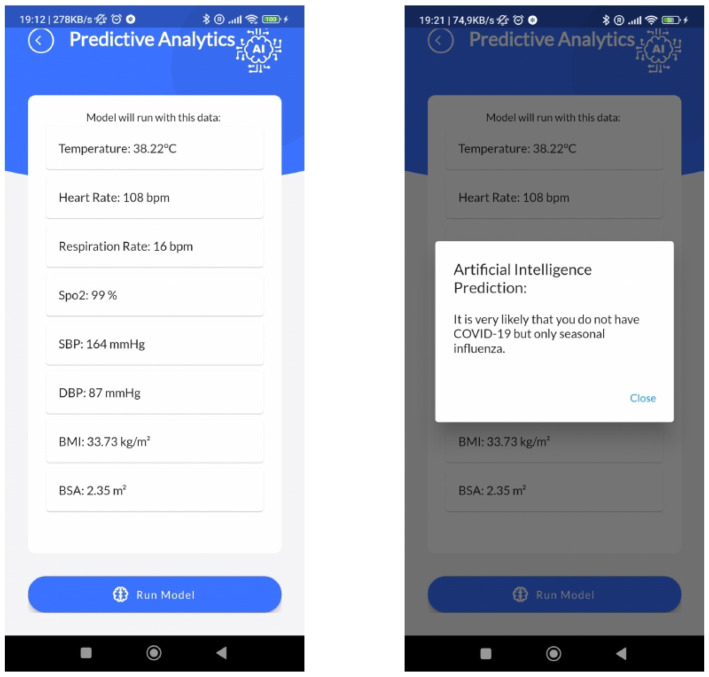
(ME)${}^{2}$ app AI predictive analytics section; in particular, the AI model used to determine the difference between influenza vs. COVID-19 symptoms is shown.

**Figure 7 sensors-22-06365-f007:**
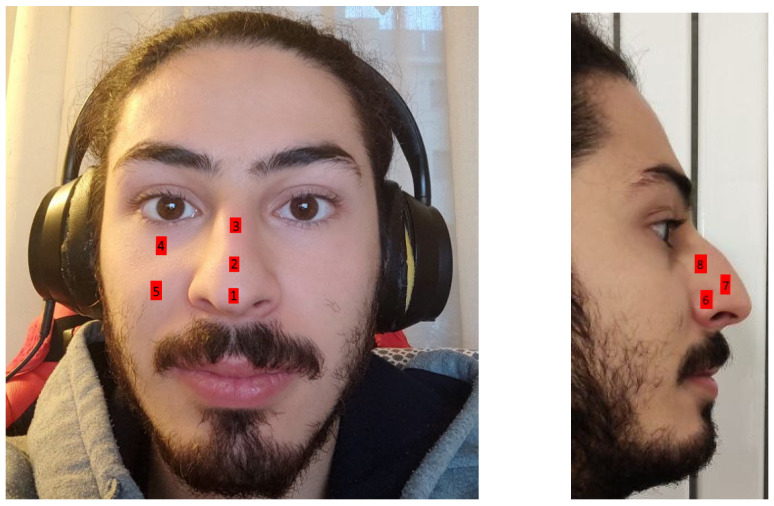
Tests of MAX30102 sensor to find the best point on the face to capture SpO${}_{2}$ and HR.

**Table 1 sensors-22-06365-t001:** Smart Mask in comparison to other solutions. x indicates that the prototype has the corresponding feature.

Name	3D-Printable	Customizable	Eco-Sustainable	Health Sensing	Other Sensing/Electronics	Display	Mobile App	AI Analytics
(ME)${}^{2}$ innovative mask	x	x	x	x	x	x	x	x
Smart Mask [14]	x		x	x	x		x	x
FaceBit [15]				x	x		x	
ADAPT [16]	x		x		x			
Lightweight and zero-power face mask [17]					x			
Face mask with heat flux sensor [18]	x (partially)				x			
AG47-SmartMask [19]	x		x	x	x			
Masquare [20]	x (partially)			x	x			
Mask for respiratory rate monitoring [21]				x			x	

**Table 2 sensors-22-06365-t002:** The table shows the absolute error from the ground-truth for each face point of both SpO${}_{2}$ (in perentage points) and HR (in bpm).

	Point 1	Point 2	Point 3	Point 4	Point 5	Point 6	Point 7	Point 8
SpO${}_{2}$	±0.1	±0.3	±0.4	±0.3	±0.2	±0.2	±0.1	±0.2
HR	±2.4	±3.5	±4.1	±3.0	±2.5	±2.8	±2.0	±2.5

**Table 3 sensors-22-06365-t003:** Performance metrics of XGBoost model for predicting the given record as COVID-19-positive or influenza.

Performance Metrics	COVID-19 Positive vs. Influenza
Accuracy	94.80%
Precision	93.00%
Recall	92.40%
Specificity	96.40%
AUC	98.60%
F1 score	92.70%

## Data Availability

Not applicable.

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
