# Peer review of "Introducing the Monitoring Equipment Mask Environment"

_sensors, 2022, doi:10.3390/s22176365_

Round 1
Reviewer 1 Report
This work proposes an intelligent mask for both user protection and health monitoring. The novelty and the originality of this work is very high.
The paper is well written and it is very interesting. However, I would organize the text in a standard way:
- I would move Table 3 and the comparison with the state-of-the-art mask in the related work section
-I would separate the project design from the experiments (e.g. Table 1 should be moved in the experimental part, where authors firstly test the measurements performances, and then they evaluate the 'intelligence' in the mask)
About this, some minor comments:
- is the distance error in Table 1 the absolute error? If so, please use this term that is more specific. Moreover, please check on your pulse-oximeter the accepted measurements error. It is usually +-2% for Sp02, if this is confirmed, it should be mentioned in the paper, since it means that almost all the Sp02 measurements agree
- Please detail how the synthetic dataset has been obtained, the data dimensionality and the class distribution
Author Response
Dear Reviewer,
thank you for your support.
As you suggested, we moved Table 3 (now Table 1) and the comparison with the state-of-the-art mask in the background section (now Background and Related Work section).
We also separated the project design from the experiments: we moved Table 1 (now Table 2) in the experimental part, where we firstly reported the measurements performances and then we evaluated the intelligence in the mask (the AI performance evaluation table is now Table 3).
Regarding the minor comments, we also addressed them. We specificated the accepted measurements error of the adopted pulse-oximeter, which actually is +-2% for both heart rate and SpO2. We also detailed how the synthetic dataset has been obtained, the data dimensionality (150x10) and the class distribution (47% covid-19 positive and 53% influenza).
We want to thank you again for the precious suggestion that we are sure will improve the quality of our work.
Reviewer 2 Report
The paper presents a mask device with new functions and is overall well structutred. I wondered whether the performance in table 2 can be compared with other similar devices.
Author Response
Dear reviewer,
we want to thank you for your great contribution.
We answer to the reviewer question asserting that the performance in table 2 (now Table 3 in the new paper version) can be compared with other similar devices if they are able to capture health vital signs and to process such data on edge with the AI model that we exploited to prove the feasibility of our solution.
Except for negligible corrections due to transpilation, we can assume that the performances are comparable with those obtained for any similar device (if there exists)